# Exploration of Sugar and Starch Metabolic Pathway Crucial for Pollen Fertility in Rice

**DOI:** 10.3390/ijms232214091

**Published:** 2022-11-15

**Authors:** Sang-Kyu Lee, Juho Lee, Mingyu Jo, Jong-Seong Jeon

**Affiliations:** 1Division of Life Science, Plant Molecular Biology and Biotechnology Research Center, Gyeongsang National University, Jinju 52828, Republic of Korea; 2Graduate School of Green-Bio Science and Crop Biotech Institute, Kyung Hee University, Yongin 17104, Republic of Korea

**Keywords:** apoplast, male sterility, *Oryza sativa*, pollen, sucrose, starch, transport

## Abstract

Starch is the primary storage carbohydrate in mature pollen grains in many crop plants, including rice. Impaired starch accumulation causes male sterility because of the shortage of energy and building blocks for pollen germination and pollen tube growth. Thus, starch-defective pollen is applicable for inducing male sterility and hybrid rice production. Despite the importance of pollen starch, the details of the starch biosynthesis and breakdown pathway in pollen are still largely unknown. As pollen is isolated from the maternal tissue, photoassimilate transported from leaves must pass through the apoplastic space from the anther to the filial pollen, where it is stored as starch. Several sugar transporters and enzymes are involved in this process, but many are still unknown. Thus, the current review provides possible scenarios for sucrose transport and metabolic pathways that lead to starch biosynthesis and breakdown in rice pollen.

## 1. Introduction

Starch biosynthesis and breakdown has been widely studied in leaves [1] and endosperm [2] in a number of plant species. However, starch biosynthesis and breakdown in pollen is still largely unknown, especially at the late phase of pollen development. During anther development, after meiosis II, four haploid microspores are generated in one tetrad. The tetrad appears to still be attached to the tapetum (Figure 1, Stage 8). The callose wall is disassembled completely, and the microspores are released from the tetrad (Figure 1, Stage 9). As a result, the microspores must feed themselves via the apoplast. Uninucleate microspore becomes more vacuolated with a round shape (Figure 1, Stage 10), in which the bulky vacuoles lead to the formation of polarized microspores, which play an important role in the asymmetric mitotic division. In the bicellular pollen, as starch accumulates inside, the vacuole diminishes gradually (Figure 1, Stage 11). The generative cell in the pollen undergoes the second mitosis and divides into two sperm cells, which generates mature tricellular pollen. The vacuole completely disappears, and a large number of starch granules form in the mature pollen grain (Figure 1, Stage 12) [3]. However, some plants, such as Arabidopsis, lack starch granules and accumulate lipid bodies, and have a distinct gene expression profile after the bicellular pollen stage [4].

Starch is the primary storage carbohydrate in mature pollen grains in many plants, including the world’s major cereal crops, barley [5], maize [6], rice [7], and wheat [8]. Pollen viability can be measured by staining starch with Lugol’s iodine solution in these plant species [9]. The production of robust pollen that accumulates sufficient starch is critical for yielding fully fertile seeds. Conversely, pollen that accumulates insufficient starch or is starchless is male-sterile and used for producing hybrid plants. An abnormality of the starch breakdown pathway also results in sterile pollen. Thus, understanding starch biosynthesis and breakdown in pollen is important for producing either robust or sterile pollen in crop species. Mutation research in pollen is the most appropriate to address this question but is challenged by the male-sterile trait caused by disrupted starch metabolism that cannot transmit mutant alleles via pollen during sexual hybridization, which does not allow us to obtain homozygous mutant progeny.

Recently, it has become possible to obtain homozygous mutant plants efficiently and rapidly at a primary generation in the model crop species rice by *Agrobacterium*-mediated transformation combined with the CRISPR/Cas system as an alternative strategy for anther culture, which requires anther tissue culture and a plant regeneration process. This new system enabled us to produce homozygous pollen mutants for many starch metabolism genes whose mutations cause male-sterile phenotypes. This review summarizes the current understanding of the sugar and starch metabolism of rice pollen based on results from mutant studies and proposes future works to fill in known gaps in the sugar and starch metabolic pathway that have not yet been clearly defined.

## 2. Sugar Transport from Maternal Anthers to Filial Pollen in Rice

During the late phase of male gametogenesis, the starch begins to accumulate at this stage to become mature pollen [3]. Because pollen is isolated from the maternal anther wall, photoassimilate transported from leaves must pass through the apoplast to move into the pollen. Imported sugars are metabolized and stored as a form of starch at the mature stage of pollen development in rice pollen. As the first step in this process, sucrose is exported to the apoplastic space of the pollen sac via the function of two sugar efflux transporters, OsSWEET11a (Sugar Will Eventually be Exported Transporter 11a) and OsSWEET11b (Figure 2). Consistent with the suggested function, their expression is abundant in the anther peduncle and vein. The *ossweet11a/11b* double mutants cause starchless pollen in rice and male sterility [10]. This sucrose transport path into the apoplastic space of the pollen sac via the SWEET transporter was also confirmed by the observation of *Arabidopsis* sucrose transporter SWEET13/14 mutants [11]. At the stage, when the flower fully opens and pollen dehiscence occurs, AtSWEET13 and AtSWEET14 are expressed in the anther, not pollen grains. The *atsweet13/14* mutant disrupts pollen maturation due to the lack of sucrose supply from the maternal tissue.

A sucrose/H^+^ symporter (Sucrose Transporter; SUT) may function in the process of sucrose transport to the pollen from the apoplastic space (Figure 2). To date, none of the SUT mutants have reported such a starchless pollen phenotype in any plant species except the *CsSUT1-RNAi* cucumber plant [12]. The male flowers of the *CsSUT1-RNAi* plants exhibited retarded growth and caused pollen growth defects, most likely due to the limitation of sugar supply into the flower organ. Of five OsSUTs in rice [13], OsSUT3 (*LOC_Os10g26470*) is highly expressed in pollen (Figure 3), but its mutant pollen has not yet been characterized in detail. In our study, mutations in OsSUT1 (*LOC_Os03g07480*) caused complete male sterility [14]. However, *ossut1* mutant pollen accumulated a normal level of starch but revealed a defect in its germination. OsSUT1 begins to express immediately after pollination and maintains its level until seed endosperm development [15]. These results indicate that OsSUT1 has a function distinct from starch biosynthesis. The CRISPR/Cas system should help create single and multigene mutants of SUT, which can provide a chance to examine those mutants and determine whether any SUT genes are involved in the transport process of sucrose from the apoplast of the pollen sac to the pollen grain.

It is unclear whether apoplastic sucrose in the pollen sac is imported into pollen grains or is cleaved into hexose, glucose, and fructose by cell wall invertase (CIN) before being imported. In principle, the CIN, which is present in the apoplastic space of the anther wall and pollen sac, can hydrolyze sucrose into hexoses there (Figure 2). The publicly available RNA-seq results suggest that several isoforms, OsCIN1 (*LOC_Os02g33110*), OsCIN2 (*LOC_Os04g33740*), OsCIN3 (*LOC_Os04g33720*), and OsCIN7 (*LOC_Os09g08072*), are expressed abundantly in anthers, but none of them are significantly expressed in pollen (Figure 3). Therefore, in rice, it is most likely that CINs are present in the cell walls of the maternal anther wall. It has previously been demonstrated in tobacco and *Arabidopsis* that CIN activity is critical for male fertility. As a result, reduced expression of anther-specific CIN genes caused male sterility [16]. In rice, the cold-inducible expression of OsINV4 (OsCIN3) in a cold-tolerant cultivar, R31, conferred normal fertility under cold conditions [17], suggesting an important role of CIN in the path of sucrose transport into rice pollen.

This result leads to the notion that sucrose unloaded from the maternal anther wall is hydrolyzed into hexose by CINs at the apoplast. The function of the hexose transporter is thus necessary for loading hexose into pollen (Figure 2). There are 28 monosaccharide transporter genes of the Sugar Transporter Protein (STP) family in rice (Figure 3) [18]. Notably, the expression levels of OsSTP7 (*LOC_Os01g38680*) and OsSTP8 (*LOC_Os01g38670*) were not decreased in the cold-tolerant cultivar R31 under cold conditions [19]. It suggests that the OsSTPs may be involved in importing hexose into pollen. However, none of the mutants of the hexose transporters resulted in male sterility. A T-DNA insertion mutant of *Arabidopsis* AtSTP9 (*At1g50310*), a monosaccharide transporter that is expressed specifically in pollen, was not male-sterile [20]. A group of SWEET efflux transporters is also involved in the capability of transporting hexose into seeds in maize and rice. Mutants of the hexose-transporting SWEETs, a transposon insertional maize mutant, *zmsweet4c*, and a TALEN-mediated rice mutant, *ossweet4*, showed a predominant role in importing sugar into seed endosperm [21]. Thus, we cannot exclude the possibility that such SWEETs may transport sugar into pollen as an alternative for the STP transporter.

## 3. Sugar Metabolic Process in Rice Pollen

Sucrose that is imported into pollen must be converted to hexose to be further metabolized for starch biosynthesis. There are two possible pathways of sucrose cleavage. One is the hydrolysis of sucrose by cytosolic neutral invertase (NIN) to glucose and fructose. The other is the production of fructose and uridine diphosphate glucose (UDP-Glc) by a glycosyl transferase, sucrose synthase (SUS) (Figure 2). There are eight neutral invertases in rice (OsNINs) [22]. OsNIN1 (*LOC_Os03g20020*) and OsNIN2 (*LOC_Os01g22900*) are predicted to localize in the mitochondrion, and OsNIN3 (*LOC_Os02g32730*) is predicted to localize in the plastid. OsNIN4 (*LOC_Os04g33490*), OsNIN5 (*LOC_Os02g03320*), OsNIN6 (*LOC_Os11g07440*), OsNIN7 (*LOC_Os04g35280*), and OsNIN8 (*LOC_Os02g34560*) are likely to localize in the cytosol (http://rice.uga.edu). Notably, of these putative cytosolic NINs, only OsNIN6 is moderately expressed in pollen, and the rest of them do not appear to be expressed (Figure 3). Although sugar metabolism following sucrose degradation into hexose is critical in the sink organ, there have been few functional studies of NIN to hydrolyze sucrose, particularly in rice. The *Arabidopsis* genome contains six SUS isoforms. The *sus1/sus2/sus3/sus4* quadruple mutant and the *sus5/sus6* double mutant did not cause any starchless phenotype in their leaves or pollen sterility [23]. None of the seven SUSs in rice were highly expressed in pollen (Figure 3). To date, no mutant studies for these OsSUSs have been reported in rice.

UDP-Glc, produced by SUS using sucrose as a substrate, can be metabolized reversibly to Glc-1-P by UDP-Glc pyrophosphorylase (UGP) in the direction of starch biosynthesis. There are two isoforms of UGP in rice. Loss-of-function mutants of OsUGP1 (*LOC_Os09g38030*) and OsUGP2 (*LOC_Os02g02560*) cause male sterility by inhibiting early pollen development [24,25] and starch biosynthesis [26], respectively. It is unclear what the exact role of OsUGP2 is during starch biosynthesis in pollen. Both UGP and SUS, which function in reversible reactions, can supply UDP-Glc in pollen, but the mutant phenotype of OsUGP is not complemented by SUS. This suggests that OsUGP2 may function in the direction of starch biosynthesis. It remains to be determined which type of sucrose cleavage enzymes primarily function during rice pollen development. Nonetheless, it is clear that sucrose must be hydrolyzed into hexose for further metabolism in pollen. Thus, mutant studies on the roles of sugar transporters, SWEET, SUT, and STP, and sucrose-cleaving enzymes, CIN, NIN, and SUS, should help determine the import route of sucrose from the pollen sac to pollen in rice.

## 4. Starch Biosynthesis in Amyloplast of Rice Pollen

It is widely known that ADP-glucose (ADP-Glc) is the building block of starch. ADG-Glc pyrophosphorylase (AGP) produces ADP-Glc and pyrophosphate (PPi) from glucose-1-phosphate (Glc-1-P) and ATP. Glc-1-P is interconverted from glucose-6-phosphate (Glc-6-P) by phosphoglucomutase (PGM) [1,2]. The rice genome has two PGMs, a cytosolic PGM (OscPGM; *LOC_Os03g50480*) and a plastidic PGM (OspPGM; *LOC_Os10g11140*). Our T-DNA insertional homozygous null mutant of OspPGM, produced by anther culture, revealed a male-sterile phenotype. The *osppgm* homozygous mutants never produced any homozygous seeds for several generations through ratoon culture. This result clearly showed that pollen defective in starch caused complete male sterility, demonstrating a critical role of the OspPGM isoform in starch biosynthesis in rice pollen (Figure 2) [27]. However, mutant plant growth was not retarded during vegetative growth, even though only a small amount of starch was present in the mutant leaf. Notably, *Arabidopsis* pPGM mutant plants are not male-sterile [28,29] because the starch level of *Arabidopsis* pollen is fairly low and not a major storage reserve.

There are six AGP isoforms in rice [30]. Each isoform is expressed in specific organs. Among them, AGP large subunit 4 (OsAGPL4) is expressed specifically in pollen. Its expression is the highest at the mature stage of pollen development, which is consistent with the level of starch accumulation. Our previous study revealed that the cytosolic AGP large subunit AGPL2 mainly functions in rice seed endosperm. Notably, AGPL4 is localized in the plastid of pollen. The loss-of-function mutant pollen of OsAGPL4 had approximately 65% starch of wild-type (WT) pollen, and the fertility of the OsAGLP4 homozygous mutant was less than 20%. These findings clearly showed that a reduced level of starch affects normal fertility, but pollen carrying a still-considerable amount of starch can successfully fertilize the homozygous mutant when there is no competition with normal WT pollen. Therefore, mutant studies of two plastidic isoforms, OspPGM and OsAGPL4, demonstrated that the level of starch is one of the crucial factors in determining pollen fertility in rice [27]. It is known that the immediate precursor of starch biosynthesis, ADP-Glc, is produced in the cytosol in seed endosperm but in the leaf plastid in rice. Our results indicate that in rice pollen, ADP-Glc is produced from Glc-6-P imported within the plastid by the concerted functions of both pPGM and AGPL4 (Figure 2).

Hexose is phosphorylated by hexokinase (HXK) to produce Glc-6-P in pollen (Figure 2). Our OsHXK5 mutant showed a reduced level of starch at the mature stage of pollen. Even though *oshxk5* mutant pollen contains approximately 75% of the starch found in the WT, it never succeeded in fertilization when it competed with WT pollen [31]. As Glc-6-P produced by HXK in the cytosol is converted to Glc-1-P in plastids by pPGM, it is essential that Glc-6-P is transported into plastids by a specific transporter. Rice Glc-6-P/P translocator 1 (OsGPT1) is the major Glc-6-P transporter on the plastidic envelope of pollen (Figure 2). The *osgpt1* mutant showed a starchless pollen phenotype that resembles the pPGM mutant [32]. Glc-6-P, which is imported into the plastid, is metabolized to starch by a concerted reaction of pPGM, AGP, and starch synthases (Figure 2) [33]. The functions of enzymes involved in starch synthesis, including starch synthases, starch branching enzymes, and starch debranching enzymes in pollen, are most likely similar to those of seed endosperm or leaf [34]. There have been no reports of single mutants for these genes causing a starchless phenotype, indicating their functional redundancy.

Uniquely, pollen in cereal crops, including rice, develops a massive vacuole before the starch biosynthesis stage. This suggests the vital role of the large vacuole as a storage organelle of sugars. Thus, there is the possibility that a large portion of sucrose is transiently stored in the vacuole in pollen. In this regard, whereas SUS is expressed at the early stage of pollen development, vacuole invertase (VIN), along with starch synthesis genes, is highly expressed at the late stage of pollen development in maize [35]. Among the two rice VINs, OsVIN2 (*LOC_Os02g01590*) is highly expressed, and OsVIN1 (*LOC_Os04g45290*) is moderately expressed in pollen (Figure 3). Previously, the *osvin2* mutant exhibited a small seed phenotype but did not show male sterility or a starchless pollen phenotype, possibly due to functional redundancy [36,37]. It remains to be determined whether a large amount of sucrose is transiently stored in the vacuole, hydrolyzed to hexose by VIN, and then exported to the cytosol in the vacuole (Figure 2). Analysis of a double mutant of OsVIN1 and OsVIN2 is necessary to address this important question. Exported sugar is subsequently phosphorylated by HXK, followed by the starch biosynthesis pathway in pollen.

## 5. Starch Degradation and Utilization in Rice Pollen Grain

Accumulated starch at the late stage of pollen development is quickly hydrolyzed soon after pollen begins to germinate the pollen tube. Given the critical role of starch accumulation in fertilization, it is not surprising that sugar released from starch is used in pollen tube development, which requires a lot of energy and building blocks to elongate the pollen tube. Glucan water dikinase (GWD) is an essential enzyme that plays a pivotal role in the first step of starch degradation. In tomato, pollen germination is impaired in the GWD mutant *legwd*. Impaired starch degradation results in an insufficient sugar supply for pollen germination and pollen tube growth [38]. The pathway of starch degradation in pollen is completely unknown not only in rice but also in Arabidopsis. In *Arabidopsis* leaf chloroplasts, β-amylase (BAM) breaks down starch into maltose [39], which is then transported to the cytosol via the maltose transporter, MEX1 (Maltose Excess 1) [40]. There are ten putative BAM genes in rice. OsBAM1 (*LOC_Os07g35940*) is predicted to be present in the cytosol because it lacks a transit peptide. OsBAM7 (*LOC_Os07g47120*) and OsBAM9 (*LOC_Os03g22790*) are likely non-functional because the active site is poorly conserved. Transcripts of OsBAM2 (*LOC_Os10g32810*), OsBAM3 (*LOC_Os03g04770*), and OsBAM8 (*LOC_Os09g39570*) are abundant in pollen (Figure 3). However, it has not yet been addressed whether mutants of these genes exhibit any defect in the fertility of rice pollen. Despite extensive research into the BAM in *Arabidopsis*, no reports of BAM multiple mutants exhibiting a male-sterile phenotype have been released either in *Arabidopsis* or rice. This may be due to the functional redundancy of several BAMs found in pollen plastids, or it could be because starch degradation by BAMs is not the primary pathway for the starch breakdown in pollen. Starch phosphorylation and dephosphorylation are also important for initiating starch degradation. Starch phosphorylating enzymes, GWD and phosphoglucan water dikinase (PWD), and starch dephosphorylating enzymes, SEX4 (Starch Excess 4) and Like Sex Four2 (LSF2) are well characterized in *Arabidopsis* leaves [41]. *Arabidopsis* mutations for these genes do not cause abnormal pollen phenotypes. In rice, pollen fertility appeared to be normal in the OsSEX4 mutant [42]. Nonetheless, it is valuable to determine whether other corresponding mutations in rice pollen that stores starch as a major reserve cause male sterility.

The rice maltose transporter OsMEX (*LOC_Os04g51330*) is a single-copy gene. In our study, the pale green leaves and growth retardation phenotype of *Arabidopsis mex1* was complemented by the heterologous expression of OsMEX into *mex1* [43], confirming that OsMEX is a functional maltose transporter. Investigation of the OsMEX mutant will help us determine whether starch is degraded by BAM in pollen (Figure 2). Current work on the *osmex1* mutant did not reveal a male-sterile phenotype. An alternative pathway by which maltotriose released by BAM is converted to glucose by the disproportionating enzyme (DPE) [44] and transported to the cytosol via the plastidic glucose transporter (pGlcT) [45] possibly complements a lack of OsMEX in rice pollen (Figure 2). The rice genome possesses one plastidic DPE (*LOC_Os07g43390*) and two pGlcT genes (*LOC_Os01g04190* and *LOC_Os09g23110*). The expression of OspDPE is high in rice pollen (Figure 3). The expressions of OsMEX and two OspGlcT genes are moderate, but their expression may be upregulated after pollination. Recently, the CRISPR/Cas-mediated mutant of DPE2 (*LOC_Os07g46790*), a rice cytosolic DPE2, showed low fertility but not complete male sterility. This is primarily due to the lack of energy for pollen tube growth in the *dpe2* mutant. In the mutant, maltose was accumulated, while glucose was decreased [46]. This strongly suggests the existence of a pathway in which BAM breaks down starch to maltose, and then exports it to the cytosol. Although a pathway for starch degradation by α-amylase is possible, none of the eight rice α-amylase genes are abundantly expressed in pollen. Therefore, this pathway, most likely, may not function in the starch degradation pathway in pollen. Functional analysis of OsMEX, OspDPE, and two OspGlcT mutations would be critical for understanding the starch degradation pathway in rice pollen (Figure 2).

Hexose, which is exported from the plastid, must be phosphorylated to be metabolized further. As Glc from the starch degradation is the major form of carbohydrate, HXK must phosphorylate Glc to form Glc-6-P. OsHXK5 and OsHXK10 are probably involved in this step during pollen germination. The loss of function of OsHXK5, which is highly expressed at the mature stage of pollen development, shows male sterility [31]. The fertility of the OsHXK10-RNAi plant is reduced when the expression of OsHXK10, which is expressed immediately after fertilization, is reduced [47].

In the process of pollen tube elongation, sucrose resynthesis should be noted. Sucrose phosphate synthase (SPS) is an enzyme that catalyzes the transfer of a hexosyl group from UDP-Glc to fructose 6-phosphate to form sucrose-6-phosphate. The OsSPS1 null mutant exhibits a male-sterile phenotype. Although more than 85% of pollen grains mature normally in the heterozygous OsSPS1 mutant, the proportion of germinated pollens is less than 43%, which is significantly lower than the 80% germination rate in WT [48]. It strongly suggests that sucrose is resynthesized during pollen germination, which has a crucial role in pollen tube germination. SUS may provide UDP-Glc, the building block of callose from sucrose to elongate the pollen tube. However, no SUS mutation with a cell wall or callose defect at this pollen tube elongation stage has been identified. Sucrose resynthesized may be essential for various metabolic processes in the rapidly growing pollen tube. Therefore, mutation research on sucrose resynthesis and its metabolism during pollen germination and pollen tube growth should be an important research topic.

## 6. Challenges and Perspectives

The viability of pollen in crop plant species is an important agronomic trait for commercial use, especially in F_1_ hybrid production. In cereal crops, starch metabolism in pollen is critical for the production of robust pollen and full fertilization. However, we are still far from fully understanding sugar and starch metabolism in pollen. The functions of key plastidic metabolic enzymes in the starch biosynthesis pathway, including glucose-6-P/P translocator, PGM, and AGP enzymes, have been characterized in rice pollen. However, it is not yet clear how sugar is loaded into the filial pollen isolated from the maternal anther wall. There are two types of sugar transporters, SWEET and SUT. Genetic studies of *ossweet11a/11b* [14] and *atsweet13/14* [15] support SWEETs function in the pathway of sucrose translocation into the pollen sac. The fate of apoplastic sucrose in the pollen sac is still questionable as there is no clear genetic study to explain it. Assuming that sucrose is cleaved by CIN in the apoplastic space, a monosaccharide transporter is essential to load hexose into pollen; however, there is no genetic evidence for this relationship thus far. It is possible to hypothesize that a portion of sucrose is hydrolyzed by CIN, but the rest of the sucrose is transported directly into pollen, though no SUT responsible for this function has been reported. Notably, the expression of VINs is high in rice pollen, suggesting that a large amount of sucrose is stored in the vacuole. Therefore, it is necessary to examine by which path sucrose is stored in the vacuole of pollen.

Over 60 genetic male sterility mutations in rice have been identified and characterized, with the majority being recessive [49]. These recessive nuclear genetic male-sterile mutants have the potential to provide excellent genetic resources for commercially important hybrid seed production in crops. As starch biosynthesis is the last step of pollen maturation, the starch level is used as an indicator of pollen viability. At the same time, starch is the major storage reserve for pollen tube growth. These important features make starch an appropriate target for male sterility. In fact, a procedure for inducing male sterility by degrading starch by expressing α-amylase in pollen grain has been successfully described [50]. With the development of a targeted mutagenesis technology with CRISPR/Cas in rice, it became possible to obtain homozygous individuals from regenerated primary plants, which was previously not possible for conventional genetic male sterility mutations or T-DNA insertional pollen mutants that can never bear homozygous seeds. Despite these advantages, because the regenerated plant by *Agrobacterium*-mediated transformation combined with the CRISPR/Cas system is male-sterile, obtaining a homozygous recessive offspring through self-pollination is theoretically impossible. Interestingly, homozygous male-sterile plants derived from abnormal starch synthesis can possibly be used as a maintainer of male-sterile mutants. For instance, the AGPL4 heterozygous pollen mutant lines produced no AGPL4 homozygous progeny, whereas the AGPL4 homozygous mutations had approximately 20% fertility, and the progeny were homozygous for the AGPL4 mutation because the mutant pollen still contained a considerable amount of starch and did not compete with WT pollen [27]. The production of the homozygous progeny of male-sterile mutant also appears in the OsHXK5 mutation. OsHXK5 homozygous mutations had approximately 10% fertility [31]. Thus, the use of the CRISPR/Cas system could be greatly beneficial for producing male-sterile plants as well as the easy maintenance of maintainer plants. Since the popularization of CRISPR will lead to the creation of many high-value-added varieties, the production of male-sterile varieties is essential for their commercial use. Of further note, all male-sterile lines produced would be non-GM plants that do not harbor any transgene.

The rice starch breakdown pathway, in contrast to the well-known *Arabidopsis* starch breakdown pathway, has received little attention, even in rice leaves. This is most likely due to the fact that, unlike Arabidopsis, rice stores sucrose as a transitory storage carbohydrate rather than starch in leaves [51,52]. Therefore, in rice, the study of starch breakdown in pollen, a tissue that primarily utilizes starch during the initial growth of pollen tubes, may be more beneficial in understanding the starch breakdown pathway than studying the leaves. Furthermore, despite its importance, little is known about how sugar from starch degradation is metabolized to complete fertilization after being exported to the cytosol during pollen tube growth. The analysis of mutants for these genes to completely understand sugar and starch metabolism in rice pollen would be valuable both scientifically and commercially.

## Figures and Tables

**Figure 1 ijms-23-14091-f001:**
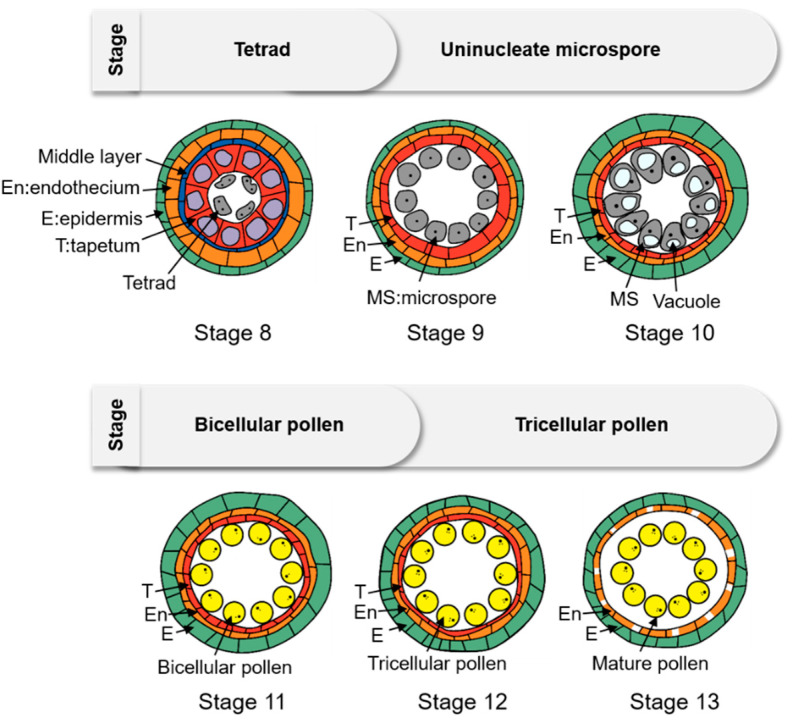
Diagrams of the late stage of rice pollen development. The number of each stage follows the description in Zhang et al. [3].

**Figure 2 ijms-23-14091-f002:**
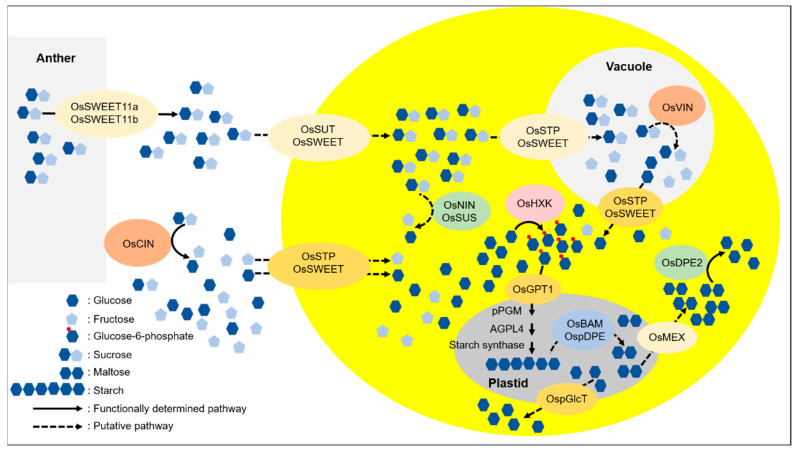
Proposed model of sugar and starch metabolic pathways in rice anther and pollen.

**Figure 3 ijms-23-14091-f003:**
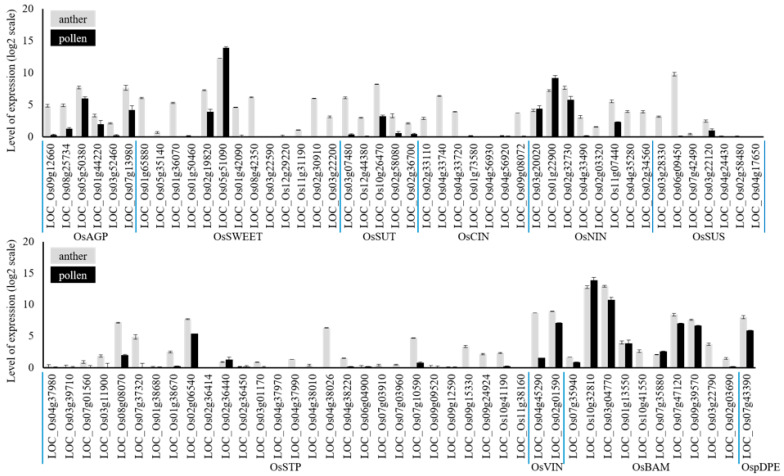
Transcript levels of sugar and starch metabolism genes in rice anther and pollen. The information was extracted from Genevestigator.

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
