# Peer review of "Exploration of Sugar and Starch Metabolic Pathway Crucial for Pollen Fertility in Rice"

_ijms, 2022, doi:10.3390/ijms232214091_

Round 1

Reviewer 1 Report

Dear Editor in Chief

The manuscript by Lee et al., entitled "Sugar and starch metabolism in rice pollen", explained the function of sugar and starch in rice pollen which is mainly related to the male sterility for the production of hybrid rice. However the manuscript is well-written and interesting, but I have a few questions the authors need to answer before accepting this paper.

Q1; The title of the review article is straightforward, just Sugar and starch metabolism in rice pollen. I cannot get enough information from the title that the authors want to express. Either they want to explain the Sugar and starch metabolism in rice pollen or the function or importance of Sugar and starch metabolism in rice pollen for fertilization or production of hybrid rice. So, I suggest authors provide a suitable title emphasizing the whole work you want to explain.

Q2; Moreover, the authors cited two figures (Figure 1 and Figure 2) in the manuscript, but I cannot find any figure, so providing only the written information in the presence of previous findings has no sense. I cannot feel any reason how I evaluate this manuscript without any results and further information. It is suggested that authors must provide some information in the forms of figures and tables, which is suitable for the reader's interest. In this form, the manuscript has no worth and cannot get the reader's attention. Moreover, it's up to the editor that provides an opportunity for the authors to provide some information in the form of figures and tables, which will be evaluated in the next round.

Author Response

Dear Editor,

We enclose our revised review manuscript for consideration for publication in International Journal of Molecular Sciences. We thank you and both reviewers for valuable comments. We made a point-to-point response to the reviewers’ comments as described below.

Major changes to address reviewers’ concerns are as follows.

- The title was changed to ‘Exploration of sugar and starch metabolic pathway crucial for pollen fertility in rice’.

- A new Figure 1 was added to help readers better understand the late stage of pollen development when starch biosynthesis occurs mainly.

- Figure 2 and Figure 3 were improved to display the content better.

- A new section(4. Starch biosynthesis in amyloplast of rice pollen) was included.

- A detailed introduction to rice pollen development and a new result on OsDPE2 were included.

- All main changes were highlighted in red on the manuscript.

Our response to reviewer #1

The manuscript by Lee et al., entitled "Sugar and starch metabolism in rice pollen", explained the function of sugar and starch in rice pollen which is mainly related to the male sterility for the production of hybrid rice. However the manuscript is well-written and interesting, but I have a few questions the authors need to answer before accepting this paper.

Q1; The title of the review article is straightforward, just Sugar and starch metabolism in rice pollen. I cannot get enough information from the title that the authors want to express. Either they want to explain the Sugar and starch metabolism in rice pollen or the function or importance of Sugar and starch metabolism in rice pollen for fertilization or production of hybrid rice. So, I suggest authors provide a suitable title emphasizing the whole work you want to explain.

Response: we changed the title to “Exploration of sugar and starch metabolic pathway crucial for pollen fertility in rice”

Q2; Moreover, the authors cited two figures (Figure 1 and Figure 2) in the manuscript, but I cannot find any figure, so providing only the written information in the presence of previous findings has no sense. I cannot feel any reason how I evaluate this manuscript without any results and further information. It is suggested that authors must provide some information in the forms of figures and tables, which is suitable for the reader's interest. In this form, the manuscript has no worth and cannot get the reader's attention. Moreover, it's up to the editor that provides an opportunity for the authors to provide some information in the form of figures and tables, which will be evaluated in the next round.

Response: I am very sorry that you reviewed an earlier version of manuscript without figures. Now we added one more new figure (Figure 1), along with two revised figures (Figure 2 and 3). We also cited a recent report supporting our current idea in the revised manuscript. Thank you.

Reviewer 2 Report

Is the work a significant contribution to the field? The presented research presents possible scenarios for sucrose transport and metabolic pathways that lead to starch biosynthesis and degradation in rice pollen. I do not see the purpose of the paper and the research hypothesis unfortunately. The paper is mostly review in nature. It is hard for me to tell if it is innovative. I see very little of the authors' own methodology and visualization of their results-there are only two graphs of which the second one does not appeal to me maybe because of the color scheme used.  The authors state that the viability of pollen in crop plant species is an important agronomic trait for commercial use, especially in F1 hybrid production. They also state that the subject of their work is only in the exploratory stage, that the mechanism is not fully known as in the case of Arabidopsis, for example. The researchers further state that analysis of mutants for these genes to completely understand sugar and starch metabolism in rice pollen would be valuable both scientifically and commercially. For me as a reviewer, this is a conclusion that confirms that it would be appropriate to supplement the paper with more of their results.   
* Is the work well organized and comprehensively described?  
* Is the work scientifically sound and not misleading?  
* Are there appropriate and adequate references to related and previous work?  
* Is the English used correct and readable?  

Author Response

Dear Editor,

We enclose our revised review manuscript for consideration for publication in International Journal of Molecular Sciences. We thank you and both reviewers for valuable comments. We made a point-to-point response to the reviewers’ comments as described below.

Major changes to address reviewers’ concerns are as follows.

- The title was changed to ‘Exploration of sugar and starch metabolic pathway crucial for pollen fertility in rice’.

- A new Figure 1 was added to help readers better understand the late stage of pollen development when starch biosynthesis occurs mainly.

- Figure 2 and Figure 3 were improved to display the content better.

- A new section(4. Starch biosynthesis in amyloplast of rice pollen) was included.

- A detailed introduction to rice pollen development and a new result on OsDPE2 were included.

  • All main changes were highlighted in red on the manuscript.

Our response to reviewer #2

The presented research presents possible scenarios for sucrose transport and metabolic pathways that lead to starch biosynthesis and degradation in rice pollen. I do not see the purpose of the paper and the research hypothesis unfortunately. The paper is mostly review in nature. It is hard for me to tell if it is innovative. I see very little of the authors' own methodology and visualization of their results-there are only two graphs of which the second one does not appeal to me maybe because of the color scheme used.  The authors state that the viability of pollen in crop plant species is an important agronomic trait for commercial use, especially in F1 hybrid production. They also state that the subject of their work is only in the exploratory stage, that the mechanism is not fully known as in the case of Arabidopsis, for example. The researchers further state that analysis of mutants for these genes to completely understand sugar and starch metabolism in rice pollen would be valuable both scientifically and commercially. For me as a reviewer, this is a conclusion that confirms that it would be appropriate to supplement the paper with more of their results. 

Response: This is a review article which includes perspectives on challenging works to further understand carbohydrate metabolic process in mature rice pollen. Now we added a new Figure 1, which can describe the pollen development process and when starch begins to accumulate. We added the paragraph to explain the late phase of pollen development when starch biosynthesis occurs in the main text. We added the most recent article about OsDPE2, which supports our idea on the starch degradation pathway. In addition, according to your comment, we improved Figures 2 and 3. Thank you.

Round 2

Reviewer 1 Report

Dear Editor

The author's revised the manuscript very well, all changes were made according to the suggestion. I am confident, now the manuscript is suitable for acceptance in IJMS. However, I found minor mistakes the authors must solve before the manuscript's final online version. 

Figure 1; the caption of figure 1 is not correct, author said that these are the pollen development stages in rice, but the author provided the figures of pollen development in rice at the late stages, starting from stage 8; where are the stages from 1-7? So, I suggest the author change the caption of Figure 1 to "Diagrams of rice pollen development at the late stages" or "Diagrams of rice pollen development at the late stages when starch biosynthesis occurs mainly." Moreover, figure 1 is only an x-axis, which makes the presentation of the figure not very well. I suggest the author change figure 1 in both the x and y-axis (In a square figure or rectangle figure).

Figure 3; The resolution of Figure 3 is not good; hard to understand the figure and bar plots. The author must provide a new figure with good resolution. 

Author Response

Dear Editor,

We enclose our revised review manuscript for consideration for publication in International Journal of Molecular Sciences. We thank you and both reviewers for valuable comments. We made a point-to-point response to the reviewers’ comments as described below.

Our response to reviewer #1

The author's revised the manuscript very well, all changes were made according to the suggestion. I am confident, now the manuscript is suitable for acceptance in IJMS. However, I found minor mistakes the authors must solve before the manuscript's final online version. 

Figure 1; the caption of figure 1 is not correct, author said that these are the pollen development stages in rice, but the author provided the figures of pollen development in rice at the late stages, starting from stage 8; where are the stages from 1-7? So, I suggest the author change the caption of Figure 1 to "Diagrams of rice pollen development at the late stages" or "Diagrams of rice pollen development at the late stages when starch biosynthesis occurs mainly." Moreover, figure 1 is only an x-axis, which makes the presentation of the figure not very well. I suggest the author change figure 1 in both the x and y-axis (In a square figure or rectangle figure).

Response: We changed the caption of Figure 1 to “Diagrams of the late stage of rice pollen development”. We also change the figure 1 to square shape.

Figure 3; The resolution of Figure 3 is not good; hard to understand the figure and bar plots. The author must provide a new figure with good resolution. 

Response: We changed all the figures to new ones with a higher resolution. Thank you.

Reviewer 2 Report

I appreciate the significant improvement in the work. The authors in most cases addressed my comments. Therefore, I have no doubts.

Author Response

Dear Editor,

We enclose our revised review manuscript for consideration for publication in International Journal of Molecular Sciences. We thank you and both reviewers for valuable comments. We made a point-to-point response to the reviewers’ comments as described below.

Our response to reviewer #2

Thank you so much for your help in improving our manuscript